# First-in-human trial of blood–brain barrier opening in amyotrophic lateral sclerosis using MR-guided focused ultrasound

Agessandro Abrahao [1,2,3,11]*, Ying Meng[2,3,4,11], Maheleth Llinas[3,5], Yuexi Huang[5], Clement Hamani[2,3,4], Todd Mainprize[4], Isabelle Aubert [2,6], Chinthaka Heyn[7,8], Sandra E. Black [1,2,5,9], Kullervo Hynynen[5,8,9,10], Nir Lipsman[2,3,4,12] & Lorne Zinman[1,2,12]

MR-guided focused ultrasound (MRgFUS) is an emerging technology that can accurately and transiently permeabilize the blood-brain barrier (BBB) for targeted drug delivery to the central nervous system. We conducted a single-arm, first-in-human trial to investigate the safety and feasibility of MRgFUS-induced BBB opening in eloquent primary motor cortex in four volunteers with amyotrophic lateral sclerosis (ALS). Here, we show successful BBB opening using MRgFUS as demonstrated by gadolinium leakage at the target site immediately after sonication in all subjects, which normalized 24 hours later. The procedure was well-tolerated with no serious clinical, radiologic or electroencephalographic adverse events. This study demonstrates that non-invasive BBB permeabilization over the motor cortex using MRgFUS is safe, feasible, and reversible in ALS subjects. In future, MRgFUS can be coupled with promising therapeutics providing a targeted delivery platform in ALS.

[1] Division of Neurology, Department of Medicine, Sunnybrook Health Sciences Centre, University of Toronto, Toronto, ON M4N 3M5, Canada. [2] Hurvitz Brain Sciences Research Program, Sunnybrook Research Institute, Sunnybrook Health Sciences Centre, University of Toronto, Toronto, ON M4N 3M5, Canada. [3] Harquail Centre for Neuromodulation, Sunnybrook Research Institute, Toronto, ON M4N 3M5, Canada. [4] Division of Neurosurgery, Sunnybrook Health Sciences Centre, University of Toronto, Toronto, ON M4N 3M5, Canada. [5] Sunnybrook Research Institute, Sunnybrook Health Sciences Centre, University of Toronto, Toronto, ON M4N 3M5, Canada. [6] Department of Laboratory Medicine and Pathobiology, University of Toronto, Toronto, ON M5S 3H7, Canada. [7] Department of Medical Imaging, Sunnybrook Health Sciences Centre, University of Toronto, Toronto, ON M4N 3M5, Canada. [8] Odette Cancer Research, Sunnybrook Research Institute, Sunnybrook Health Sciences Centre, University of Toronto, Toronto, ON M4N 3M5, Canada. [9] Institute of Biomaterials and Biomedical Engineering, University of Toronto, Toronto, ON M5S 3H7, Canada. [10] Department of Medical Biophysics, University of Toronto, Toronto, ON M5S 3H7, Canada. [11] These authors contributed equally: Agessandro Abrahao, Ying Meng. [12] These authors jointly supervised this work: Nir Lipsman, Lorne Zinman. *email: agessandro.abrahao@sunnybrook.ca

Amyotrophic lateral sclerosis (ALS) remains a fatal neurodegenerative disease characterized by the progressive degeneration of both the upper and lower motor neurons (UMNs and LMNs). There is an incomplete understanding of the disease pathophysiology and current interventions only mildly slow disease progression. While preclinical studies in ALS rodent models have primarily focused on the distal-to-proximal motor neuron dying-back hypothesis[1], recent data support a key pathological role of UMN in the motor cortex[2]. Signs of cortical neuronal and glial dysfunction have been observed early in the disease course and preceded spinal cord neuronal degeneration in ALS rodent models[3,4], supporting a motor cortex driven dying-forward hypothesis[2]. Early cortical motor neuron dysfunction in ALS has also been demonstrated in human studies using transcranial magnetic stimulation, neuroimaging and electrophysiological measures[2,5,6]. Therefore, interventions that target cortical motor neurons alone or in combination with LMN-directed therapies may be essential in slowing or halting disease progression.

Access to the neurons and glia of the primary motor cortex remains challenging as the blood–brain barrier (BBB) limits the transfer of therapeutics from the intravascular compartment. While there is preclinical and post-mortem evidence of abnormal BBB ultrastructure in ALS[7,8], the BBB still restricts the passage of drugs, antibodies, gene carriers and stem cells from the capillaries to the central nervous system (CNS)[9]. Even riluzole, an approved drug for ALS, has limited availability in the brain tissue given BBB-related drug efflux mechanisms[10].

Presently, the targeted delivery of biotherapeutics to the brain requires an invasive neurosurgical procedure, which can have serious complications. Promising biotherapeutic approaches to UMN pathology in ALS, including neural progenitor cells[11] and adeno-associated virus (AAV) expressing small-hairpin RNA to suppress mutant superoxide dismutase-1 (SOD1)[3], have been surgically injected into the motor cortex of ALS rodents, resulting in significantly increased lifespan. Stereotactic implantation of autologous CD133+ stem cells into the primary motor cortex has also been investigated in a small cohort of ALS subjects with limited success[12]. Alternatively, non-invasive delivery methods across the BBB via facilitated transcellular transport (e.g., drug modifications with transferrin or insulin receptors[13]), drug efflux inhibition (e.g., elacridar[10]), or diffuse BBB breakdown (e.g., mannitol) lead to widespread CNS uptake and potential off-target effects. Given the limitations of open surgery and non-targeted pharmacological delivery approaches, the development of a non-invasive, image-guided method to safely and temporarily open the BBB for targeted delivery would represent a paradigm shift in the treatment of ALS.

Transcranial magnetic resonance-guided focused ultrasound (MRgFUS) combined with intravenous ultrasound contrast (perflutren lipid microbubbles) has emerged as an incision-less technique to transiently open the BBB in targeted CNS regions[14]. The MRgFUS device is comprised of a multi-element phased array transducer system that can target virtually any brain region with millimetric accuracy, using real-time MRI feedback for tissue monitoring and intraoperative target guidance[15]. At higher frequencies, MRgFUS thalamotomy was found to be effective in treating medically refractory essential tremor in a randomized controlled trial[16]. At lower powers, ultrasound interacts with injected microbubbles resulting in transient mechanical disruption of the BBB without thermal injury[17–19]. This has been shown to enhance drug delivery to multiple brain regions in small-to-large animals, including cortical targets[20], the hippocampus[21], striatum[21,22], and brainstem[23]. Although safe BBB opening in eloquent and non-eloquent brain regions has been demonstrated in subjects with high-grade gliomas using surgically implanted intracranial pulsed ultrasound device[24,25], transcranial MRgFUS BBB opening is a non-invasive approach and has optimal spatial resolution for discrete and precise BBB targeting in eloquent regions, such as the primary motor cortex. In recent pilot clinical trials, non-invasive MRgFUS BBB opening in small and non-eloquent regions in and around gliomas[26] and in the frontal lobe of patients with Alzheimer's disease (AD)[14] was safe and transient.

This study represents the first-in-human trial testing the feasibility and safety of transient permeabilization of the BBB using MRgFUS in subjects with ALS. It is also the first attempt to apply this technology to eloquent cortical and subcortical tissue in human subjects as primary motor cortex was targeted using functional MRI (fMRI). Here, we show safe and reversible motor cortex BBB opening using MRgFUS at the target site immediately after sonication in all subjects without serious adverse events.

## Results

**Participants and target ascertainment.** Four eligible subjects, two women and two men with median age 61 years (range: 56–70), diagnosed with ALS (probable El Escorial criteria) and clinical evidence of upper motor neuron dysfunction were enrolled in the study (Fig. 1). Table 1 lists their baseline demographics. Each subject's sonication targets (Supplementary Fig. 1) were individualized according to arm (Fig. 2a) or leg (Fig. 3a) fMRI activation in the primary motor cortex to ascertain the homunculus topography. Hand activation by subject two and three was spatially shifted laterally from what is typically described as the hand knob area of the precentral gyrus. Cortical atrophy of the hand functional region was also noted on visual inspection at baseline in subject three, who presented with arm-onset phenotype. Subject four, who had leg-onset ALS, revealed bilateral motor cortex activation on fMRI.

**Outcomes**. MRgFUS-mediated opening of the BBB was successful in all four subjects as judged by demonstration of gadolinium leakage directly at the sonication target (Figs. 2, 3) and further indicated by increases in normalized intensity ratio (IR) on MRI (Fig. 4). Notably, the IRs decreased one day following the procedure indicating reversible BBB permeability. The total target volume of $350\,mm^3$ was permeabilized with 2–4 rounds of sonication with a median power of 8.0 (range: 6.0–10.0) W (Supplementary Table 1). Real-time MR thermometry monitoring did not detect temperature elevation.

BBB opening in the primary motor cortex was well tolerated with few mild-to-moderate procedure-related adverse events (AEs) (Table 2). There was no procedure- or device-related serious adverse events (SAEs) from intervention to 60 days of follow-up. Subjects tolerated the procedural time and intravenous microbubble injections, with no immediate tissue reaction such as hemorrhage or edema. Medications administered for patient's comfort and AE management during the intervention are listed in Supplementary Table 2. Median duration in the MRgFUS procedure was 66 min (range: 43–173), which included acquisition of planning sequences, targeting and sonications. Median duration of pre- and post-sonication MR scanning was 115 min (range: 90–147). No change in the neurological status was detected during the sonications in any of the patients and the 24-h inpatient observation period was uneventful. All patients were discharged on the first post-operative day. In subject four, a transient and asymptomatic hyperintense sulcal and parenchymal fluid-attenuated inversion recovery (FLAIR) signal was observed in a portion of the sonicated region on the day 1 and resolving on day 7 MR scans, without any accompanying changes on T2, T2* gradient echo (GRE), or diffusion weight imaging (DWI)

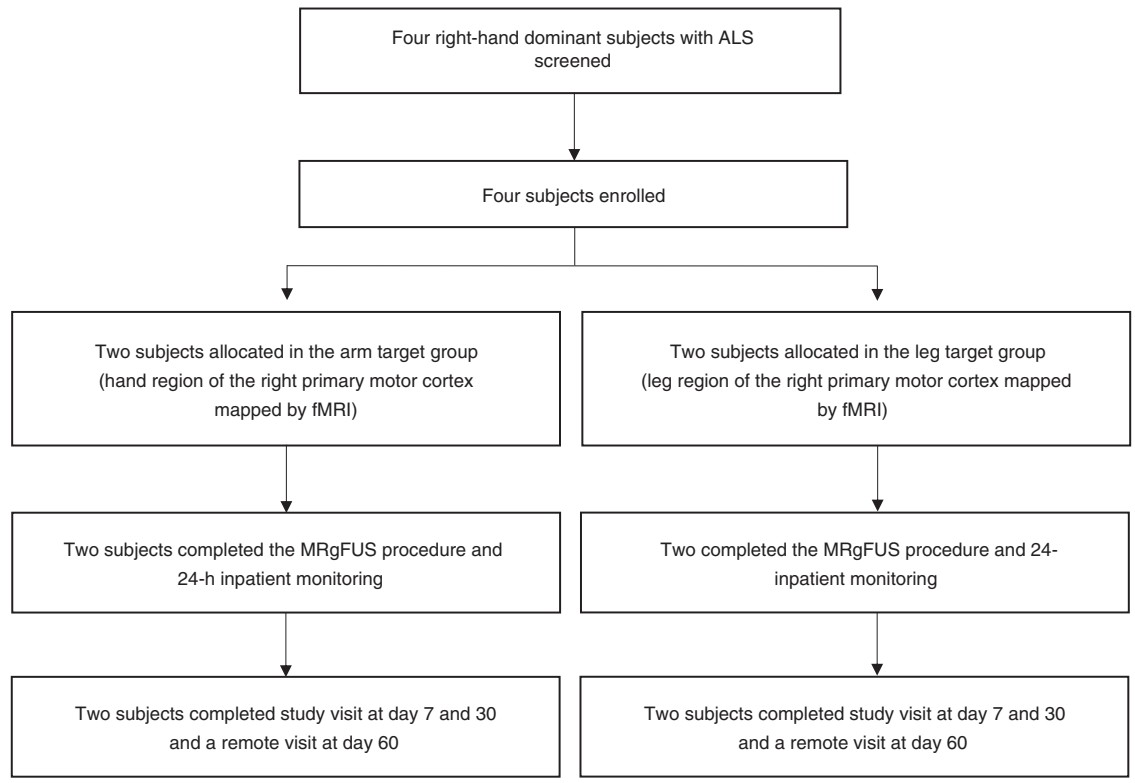

**Fig. 1** Trial profile

| | Disease duration (years) | Site of ALS onset[a] | MRC sum score: Arms[b] | MRC sum score: Legs[b] | SVC[c] | ALSFRS-R score (total 48) | Modified Ashworth score[d] | MoCA score (total 30) | Pre-existing conditions | ALS specific treatment |
|---|---|---|---|---|---|---|---|---|---|---|
| Subject 1 | 4.2 | Lower extremity | 52 | 19 | 83% | 32 | 0 | 26 | HTN, IBS, GERD, chronic pain, osteoporosis, anxiety | – |
| Subject 2 | 4.3 | Upper extremity | 22 | 47 | 57% | 30 | 0 | 30 | HTN, migraine, childhood asthma | Riluzole |
| Subject 3 | 4.5 | Upper extremity | 27 | 60 | 80% | 35 | 0 | 27 | HTN, psoriasis | Riluzole and Edaravone |
| Subject 4 | 4.1 | Lower extremity | 56 | 16 | 69% | 33 | 0 | 25 | Chronic pain | - |

**Table 1 Baseline patient characteristics**

[a]defined as first reported weakness
[b]Medical Research Council (MRC) manual muscle strength rating for bilateral shoulder extensors, elbow flexors, elbow extensors, finger extensors, thumb flexor at the interphalangeal joint, abductor of the index finger and thumb abductors, hip flexors, hip abductors, knee extensors, knee flexors, ankle dorsiflexors and plantar flexors. Normal MRC sum score is 70 for both arms and 60 for both legs
[c]Slow vital capacity (SVC) was reported as percentage of the predicted for age, sex and body mass index
[d]Zero indicates no spasticity
*ALSFRS-R* revised ALS Functional Rating Scale; Normal score equals 48, *MoCA* Montreal Cognitive Assessment; Normal score equals 30, *HTN* hypertension, *IBS* irritable bowel syndrome, *GERD* gastroesophageal reflux disease

sequences (Supplementary Fig. 2). This radiologic finding did not correlate with any new symptoms, neurological signs, or focal electroencephalography (EEG) changes and completely resolved at day-30 MRI. In all participants, follow-up MR sequences did not demonstrate parenchymal or subarachnoid hemorrhage, ischemia, gliosis, or worsened precentral gyrus cortical atrophy (Supplementary Fig. 3) up to 30 days following the intervention.

Longitudinal neurological assessments revealed no accelerated disease progression as measured by clinical assessment and Medical Research Council (MRC) scores for the limb contralateral to the sonicated cortex up to 30 days after sonications and

ALS Functional Rating Scale - Revised (ALSFRS-R) scores up to day 60 (Supplementary Table 3). Finally, there were no clinically significant longitudinal changes in cognition or spasticity as measured by Montreal Cognitive Assessment (MoCA) and Modified Ashworth scores, respectively. EEG revealed no changes from baseline and no epileptiform discharges 30 days following BBB opening. All safety laboratory testing was unremarkable.

## Discussion

Therapeutic access to degenerating neurons and glial cells in the primary motor cortex is essential in the development of disease-

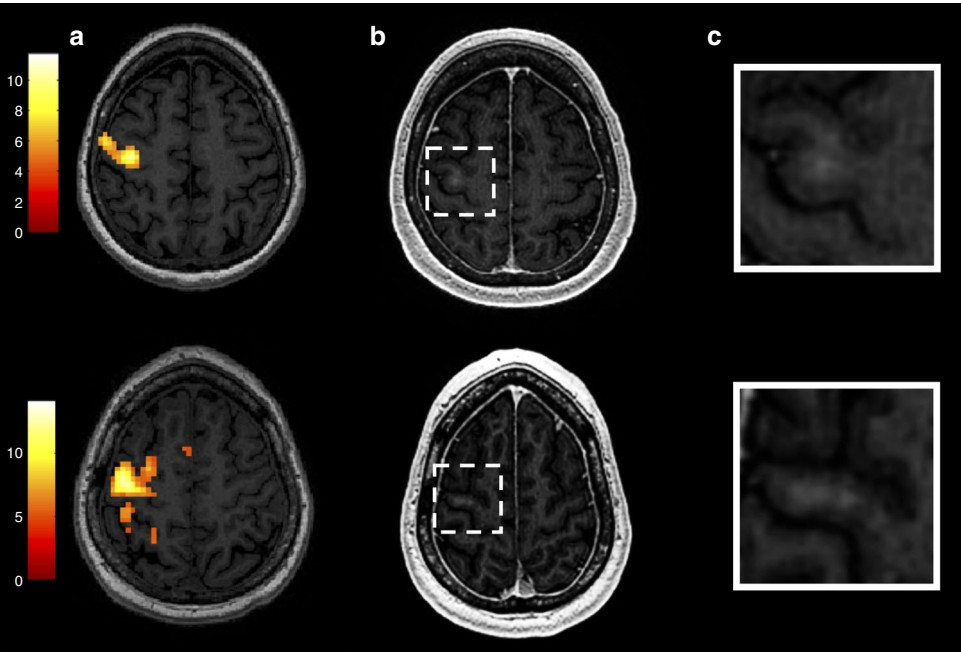

**Fig. 2** Feasibility of blood–brain barrier opening using MRgFUS in the hand control region of the right primary motor cortex. **a** Hand control region mapping using motor-task functional MRI (fMRI). **b**, **c** show confirmation of successful blood–brain barrier permeabilization in the sonicated primary motor cortex area as demonstrated by new gadolinium enhancement in T1-weighted imaging. Subject two is represented in the top row and subject three in the bottom row

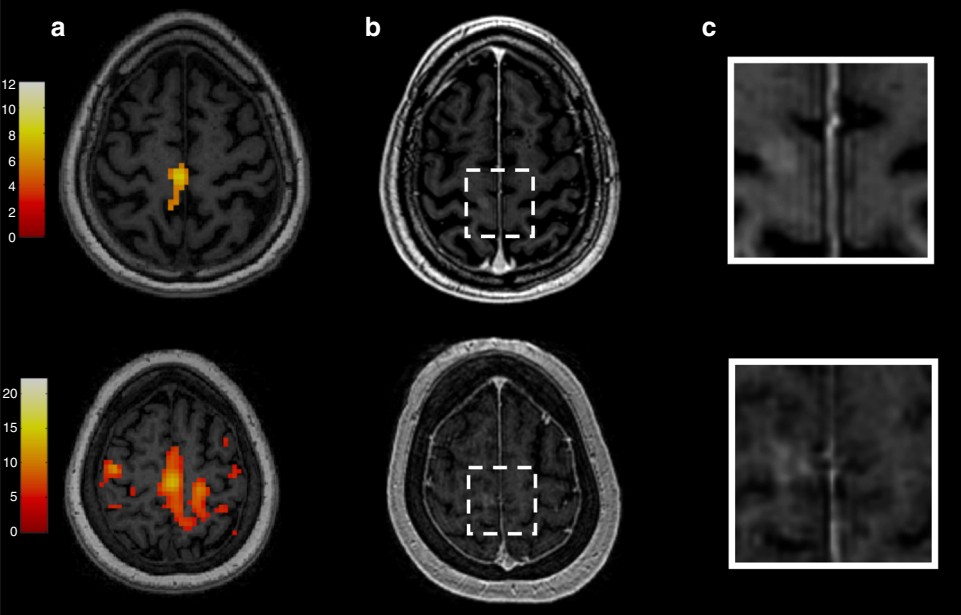

**Fig. 3** Feasibility of blood–brain barrier opening using MRgFUS in the leg control region of the right primary motor cortex. **a** Leg control region mapping using motor-task functional MRI (fMRI). **b**, **c** show confirmation of successful blood–brain barrier permeabilization in the sonicated primary motor cortex area as demonstrated by new gadolinium enhancement in T1-weighted imaging. Subject one is represented in the top row and subject four in the bottom row

modifying treatments in ALS. However, the BBB significantly restricts the penetration of numerous interventions from the intravascular compartment to the CNS. In this study, we demonstrated that the BBB can be precisely, safely, and temporarily opened over a targeted region in the primary motor cortex in ALS subjects using transcranial MRgFUS. It also

represents the early clinical application of non-invasive, tran-scranial MRgFUS to safely permeabilize fMRI-directed eloquent brain regions as prior MRgFUS studies targeted sub-cortical white matter regions in humans[14,26].

Targeting of the primary motor cortex was achieved with millimetre accuracy and tissue sonication was well tolerated by all

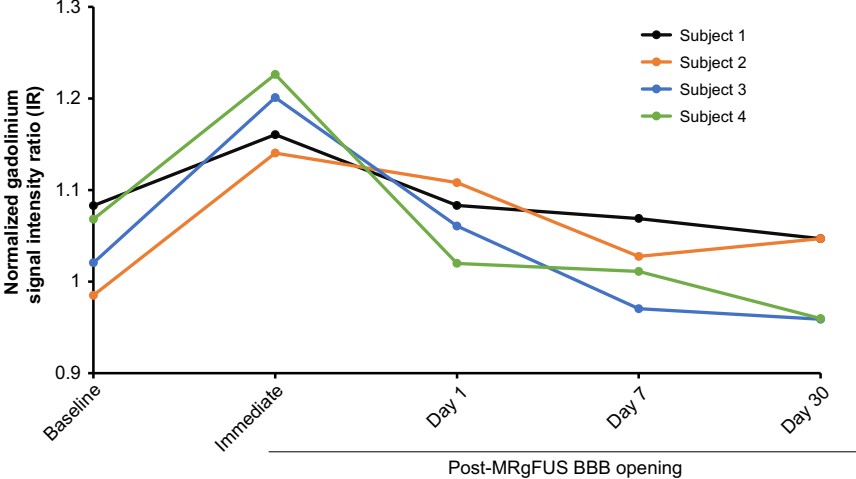

**Fig. 4** Baseline and post-procedure gadolinium signal intensity ratio (IR) within the sonicated region on T1-weighted imaging. IRs were normalized to the contralateral, unsonicated mirrored region. Increased IRs were seen immediately after blood–brain barrier opening and returned to baseline within 24 h, indicating reversible BBB permeability. Raw data are provided in the Source Data file

| Table 2 Procedure-related adverse events | | |
|---|---|---|
| **Adverse event** | **Severity** | ***N*** |
| Intraprocedural events | | |
| Headache | Moderate | 3 |
| Vagal response | Moderate | 1 |
| Scalp pain, edema, or bruising attributable to placement of the stereotactic frame | Mild | 2 |
| Musculoskeletal pain | Mild to moderate | 2 |
| Scalp petechial rash | Mild | 1 |
| Transient FLAIR hyperintensity in sonicated brain tissue (Supplementary Fig. 2) | Mild | 1 |

subjects with no detectable SAEs, hemorrhage or persistent lesions on longitudinal imaging. While GRE is sensitive to microhemorrhages, the absence of changes does not rule out trace erythrocyte extravasation which has been observed in post-mortem animal MRgFUS studies[18,27]. In these models, erythrocyte extravasation was generally limited to the perivascular spaces and the animals were asymptomatic. While there is pre-clinical and post-mortem evidence suggesting endothelial cell dysfunction and impaired BBB repair mechanisms in ALS[8], in all participants of this study, BBB permeabilization was reversible, without radiological evidence of significant inflammation or tissue damage 30 days after the procedure. One participant (subject four) developed a delayed and self-limited, small parenchymal FLAIR hyperintensity within the sonicated region. This asymptomatic finding was inconsistent with hemorrhage, ischemia, or gliosis and completely resolved on imaging obtained 30 days following the intervention. While the aetiology and clinical significance are uncertain, the finding may represent serum extravasation of gadolinium, albumin and/or immunoglobulins from BBB opening[28,29].

The complication risk of the MRgFUS BBB opening procedure is mitigated by the inclusion of a number of key technical and design safety features. First, the optimal sonication power for safe BBB disruption at each target is determined through tissue-specific ramp test, based on the acoustic and cavitation feedback

measured by the MRgFUS system[27]. Second, real-time MR thermography and acoustic spectrum monitoring, along with the patient's ability to communicate adverse symptoms, ensure target accuracy, tissue integrity and prompt recognition of any complication during sonication. These intraprocedural safety assessments are of particular importance when targeting eloquent brain regions with increased vascularity, such as the primary motor cortex. With these safeguards in place, BBB opening with MRgFUS likely poses a lower risk and faster recovery time than open neurosurgical procedures.

MRgFUS-mediated BBB opening is a non-invasive alternative to stereotactic surgery for the targeted delivery of therapeutics. Preclinical studies have demonstrated the effectiveness of MRgFUS in enhancing targeted brain bioavailability of large drugs (e.g., chemotherapeutics[23,30], antibodies[20,31–33], AAV9[21], and cells[34–36]) injected intravascularly in animal models. Multiple MRgFUS procedures are also feasible for therapeutics with repeated dosing as demonstrated in a pilot MRgFUS trial in subjects with AD[14] and preclinical studies[36,37] MRgFUS creates a window for candidate large investigational compounds, infused peripherally to target degenerating neurons and glial cells in the primary motor cortex of ALS patients. In ALS animal models, antibodies[38,39], AAVs[3] and progenitor cells[11] were efficacious in prolonging survival, but required invasive surgical delivery procedures.

Antibody-mediated reduction of misfolded or toxic proteins is a therapeutic strategy in ALS[3,39]. In *SOD1* transgenic ALS mice, antibodies binding to misfolded SOD1 were effective in prolonging survival following an intraventricular infusion[38,39]. The technical feasibility and utility of MRgFUS in delivering antibodies to sonicated tissue has been demonstrated in animal models of AD and brain tumours[20,31–33]. Humanized antibodies to key epitopes such as cortical misfolded SOD1 or targets of neuroinflammation and survival pathways implicated in ALS can be coupled with MRgFUS for brain delivery in future trials[14].

Effective application of upstream strategies to knock down *SOD1* expression using AAV9-mediated RNA therapy requires delivery by ventricular intrathecal injection or surgical implantation in the motor cortex[3,40]. In contrast to intrathecal injections, MRgFUS significantly optimized the targeted AAV9 bioavailability in sonicated brain regions with lower intravenous

viral load required[21]. Also, while AAV9 introduced via lumbo-sacral intrathecal administration can diffuse across multiple anterior horn levels of the spinal cord, its diffusion into the motor cortex was limited[41], highlighting an additional advantage of the MRgFUS-mediated brain delivery.

Finally, another promising strategy in preclinical ALS models is to enhance neurotrophic support in the motor cortex. Thomsen et al.[11] demonstrated increased neuroprotective effects of glial cell line-derived neurotrophic factor (GDNF) after stereotactic intracortical transplantation of GDNF-secreting neural pro-genitor cells, leading to significant UMN and LMN protection. MRgFUS has been safely used to deliver progenitor cells[34] and other cell types[36] to discrete brain regions in animal models following intra-arterial and intravenous cellular injection and may be a non-invasive delivery option in future studies.

To enhance a potential synergistic effect on UMNs and LMNs, targeted brain therapeutic delivery using MRgFUS can be coupled with non-targeted, lumbosacral intrathecal therapeutic injections, which have unreliable motor cortex penetration and are currently being used in ALS clinical trials. MRgFUS blood-spinal cord barrier opening has not yet been developed in humans, but has been performed safely in the spinal cord of non-ALS rodent models[42,43]. An early study demonstrated enhanced chemother-apy delivery using MRgFUS blood-spinal cord barrier opening in rodents[44]. However, the bony vertebrae of the spinal column in primates present a technical challenge for focusing the ultrasound and further validation and optimization studies are necessary before translation to humans[45].

This study represents the initial step towards the validation of a non-invasive brain delivery platform in ALS, but there are several limitations. First, the ALS cohort enrolled in this open-label, single-centre study was small with limited generalizability and power to infer AE rates with precision. For safety concerns, only patients with limb-onset ALS and severe weakness were included and only a small volume of the primary motor cortex was tar-geted. While the minimal volume of BBB opening for sufficient therapeutic distribution is unknown and likely depends on drug properties, future trials will explore the safety and feasibility of sonicating larger volumes or the entirety of the primary motor cortex, along with frontotemporal regions that are also affected in ALS with frontotemporal dementia. Current preclinical evidence support the feasibility of targeting large hemispheric volumes in canines, whose brain dimensions more closely resemble that of humans[46]. Further, the clinical MRgFUS device can achieve a larger volume of BBB opening through a summation of targets and wider spacing of spots within each target[14]. Ongoing clinical trials using focused ultrasound (NCT03616860, NCT03712293, NCT03714243, NCT03626896) as well as pulsed ultrasound (NCT03744026) are underway to treat larger tumour regions and peri-tumour tissue to enhance chemotherapy delivery.

The qualitative and semi-quantitative assessment of BBB per-meability through contrast T1-weighted MRI is also a limitation in this study. Because the peak gadolinium enhancement is expected to be immediately after BBB opening, a delay between sonications and contrast MRI acquisition due to patient transfer, head frame removal, and rest will contribute to decreased visualized IR in the sonicated target. In future MRgFUS BBB opening trials, dynamic-contrast enhanced MRI sequence and MR coil improvements can more quantitatively and accurately assess BBB permeability.

Cortical plasticity and functional reorganization of the motor homunculus topography in ALS[47,48] can limit the ascertainment of functional regions of the primary motor cortex. We found motor activation by fMRI was necessary in most of the subjects to more accurately determine the functional arm and leg-controlling cortical regions. We observed lateral shifts in the cortical hand

area, as well as overactivation of bi-hemispheric anterior struc-tures (e.g. premotor and supplementary motor area) as previously described[47–49].

In addition to providing a window for therapeutic access, BBB opening exposes the central nervous system microenvironment to circulating proteins. For instance, transient microglial activation was observed in a transgenic rodent model of AD after BBB opening and resolved at 15 days[50]. The implication of transient microglial activation in ALS is unknown as humoral and cellular mechanisms have been associated with both neuroprotection and toxicity[51]. Neuroinflammation and endothelial, neuronal and glial function should be further investigated in upcoming MRgFUS trials using advanced MR imaging and spectroscopy, cortical excitability measures (e.g., short intracortical interval parameter of threshold-tracking transcranial magnetic stimulation), PET imaging (e.g. TPSO[52]), along with serum and cerebrospinal fluid biomarkers.

In conclusion, the reversible permeabilization of the BBB in small regions of the primary motor cortex using transcranial MRgFUS with microbubbles was safe and tolerated in subjects with ALS. This study represents the seminal initial step in establishing a delivery platform whereby MRgFUS-mediated BBB opening can be coupled with the systemic administration of the most promising ALS therapeutics to directly target the degen-erating motor cortex.

## Methods

**Study design**. This prospective, open-label, single-arm, single-centre first-in-human study was designed to evaluate the technical feasibility, reversibility and safety of BBB opening in the primary motor cortex (precentral gyrus) of subjects with ALS. The non-dominant motor cortex region corresponding to a severely weakened contralateral limb was chosen. The study was approved by the Research Ethics Board at Sunnybrook Health Sciences Centre (SHSC) and Health Canada and was registered with ClinicalTrials.gov NCT03321487. All subjects provided written informed consent prior to enrolment after a detailed discussion of the study rationale, risks and the investigational nature of the procedure. This trial complied with the International Conference on Harmonization guideline for Good Clinical Practice, Tri-Council Policy Statement on ethical conduct for research involving humans (TCPS-2), and ISO 14155.

**Participants**. Four right-hand dominant participants aged ≥18 years were enrolled in the study. Each was diagnosed with laboratory-supported probable, clinically probable or definite ALS according to the revised El Escorial criteria[53], which requires evidence of UMN dysfunction in at least one body region. Participants had slow vital capacity (SVC) ≥50% for predicted age and body habitus and severe left hand or leg weakness as defined by MRC muscle strength grade ≤3 in index finger abduction and thumb abduction or hip flexors and ankle dorsiflexors, respectively. Participants on a stable dose of oral riluzole (up to 50 mg twice daily) for at least 30 days or intravenous edaravone for at least two cycles were eligible for the study. Edaravone and riluzole were not administered on the day of the MRgFUS proce-dure and restarted after discharge. Efforts were made to balance sexes in this study.

Important exclusion criteria included co-existent frontotemporal dementia, systemic or cerebral vasculopathy, auto-immune conditions, bleeding disorder or taking an anticoagulant, contraindication to MRI or gadolinium contrast, intracranial hemorrhage or prohibitive structural lesions seen on baseline MRI, American Society of Anesthesiologist classification > III, cystatin C glomerular filtration rate < 30 mL per min per 1.73 m$^2$, unstable cardiovascular or pulmonary disease, or contraindication to intravenous microbubble injection, including known or suspected cardiac shunt or perflutren hypersensitivity. Urine pregnancy test and use of contraception during the course of the study was required for women of childbearing potential. Detailed eligibility criteria are listed in Supplementary Methods.

Enrolled patients were allocated in an open-label manner to the arm ($n = 2$) or leg ($n = 2$) target group depending on which left-sided limb was weaker. In each group, MRgFUS sonications targeted the right motor homunculus region corresponding to the hand or leg and aimed to open a maximum BBB volume of 1 cm$^3$. Target accuracy was ascertained using hand or foot task activation functional MRI prior to the sonication as described below.

**Functional MRI acquisition and task**. All fMRIs were performed using the same study 3-Tesla scanner (Signa MR750; GE Healthcare, Milwaukee, Wis.) with an eight-channel head receiver coil. Anatomical images were acquired with 3D fast spoiled gradient echo (3D-FSPGR) [IR-FSPGR (inversion recovery prepared fast spoiled gradient echo)] sequence with 1 mm thickness. The scan parameters were

TE = 2.94 ms, TR = 7.65 ms, and matrix size = 265 × 265. fMRI parameters were 130 temporal volumes of 36 slices with 4 mm thickness; TR = 2000 ms, TE = 30 ms; flip angle = 70°; and matrix size = 64 × 64. The first four volumes were discarded for T1 effects. The functional activation images were acquired during a block-design motor task with total duration of 4 min 28 s. Six blocks of repetitive movements to written command were interspersed with blocks of black-on-white crosshair fixation to elicit the blood-oxygen-level-dependent hemodynamic response to motor activity. Each block lasted for ten seconds. Subjects were instructed prior to entering the scanner to perform hand squeezes or foot taps and had the opportunity to practice.

**MR-guided focused ultrasound intervention**. The MRgFUS BBB opening procedure was performed using the 220 kHz ExAblate Neuro 4000 system type 2.0 (InSightec®, Israel) with 1024 ultrasound transducers coupled to the 3-Tesla MRI. First, the subject's hair was closely shaved to the scalp. For stereotactic accuracy, each subject was fitted with a Cosman-Roberts-Wells stereotactic frame under local anaesthetic. Each subject was placed supine on the MR table with compression stockings and care was taken to pad all pressure points. The frame was then fixed to the ExAblate helmet with intervening degassed water between the scalp and transducers.

An anaesthesiologist was present for the entire procedure to monitor vital signs and to provide analgesia and conscious sedation as necessary. MR-compatible, non-invasive monitoring of vital signs and electrocardiogram was maintained throughout the procedure. The subjects remained awake and had available a stop mechanism in case of discomfort or emergency. First, planning MR sequences were acquired for registration to baseline CT scan. To achieve BBB opening within the arm or leg regions, two sonication targets of 5 × 5 × 7 mm (175 mm³ each, 350 mm³ total) were placed over the arm or leg region as mapped by task fMRI, avoiding overlap with any voxels containing sulci and visible vessels within two contiguous axial slices as to mitigate the potential risk of hemorrhage. Within each sonication target, the focused ultrasound beam was electronically steered among four spots within the target as represented in Supplementary Fig. 4A.

For transcranial focused ultrasound applications, accurate estimation of in vivo tissue pressure for individual subjects at this level of power is limited given highly variable skull geometry, density, and target locations. Therefore, the optimal power for BBB opening was determined by cavitation feedback at each target through a ramp test[14,27]. This involves applying short sonications with 5% power increments until the device's hydrophones detect sub-harmonic acoustic feedback from the target, indicating the cavitation threshold. The optimal power for BBB opening was estimated as 50% of this threshold.

The power ramp test was then followed by one to two 90 s sonication cycles until the BBB was opened, each coupled with intravenous 4 µl/kg perflutren microbubbles (Definity®, Lantheus Medical Imaging, USA). The total dose of perflutren did not exceed 20 µl/kg. During each 90 s sonication cycle, ultrasound was delivered in burst mode with pulse repetition period of 300 ms and 0.88% duty cycle per each of the four spots within a target (Supplementary Fig. 4B).

Real-time monitoring during sonications included acoustic monitoring, MR thermometry, and direct patient feedback. We were able to acquire MR thermometry without interference, using standard fast spoiled gradient echo (FSPGR) sequence with parameters TR = 26 ms and TE = 13 ms. Conversely, there is no MR interference to cavitation receivers built-in the focused ultrasound array. Subsequent to each sonication cycle, the subject was examined clinically and radiologically for any AEs, including gadolinium enhanced T1-weighted and GRE sequences to assess BBB permeability, tissue integrity and microhemorrhage. Visualization of new gadolinium enhancement in the sonicated brain parenchyma targeted signified successful BBB opening and the procedure was terminated. Subjects then returned for MRI with the head coil for high definition imaging. All subjects were admitted to the neurosurgical unit for overnight observation followed by clinical and MRI assessments for reversibility of BBB opening and AEs before discharge from hospital the next day.

**Outcomes**. The primary outcome was safety and feasibility of transient BBB opening by MRgFUS in the primary motor cortex. Safety was measured as the occurrence and severity of device- and procedure-related AEs and SAEs over the 60-day study duration. AE and SAE definitions are detailed in Supplementary Note 1 and Supplementary Table 4 and were captured by clinical assessment, laboratory testing, and neuroimaging immediately after BBB opening, and at day 1, 7, and 30 post-MRgFUS procedure and over the phone at day 60. Neurological exam included MRC manual muscle strength rating (grade 0–5) of the shoulder extensors, elbow flexors, elbow extensors, finger extensors, thumb flexor at the interphalangeal joint, abductor of the index finger, thumb abductors, hip flexors, hip abductors, knee extensors, knee flexors, ankle dorsiflexors and plantar flexors. Normal MRC sum score for the arm is 35 and for the leg is 30. Safety laboratory testing included complete blood count, C-reactive protein, erythrocyte sedimentation rate, creatinine, cystatin-C, liver enzymes and electrolytes. Safety MRI sequences included T1 with and without gadolinium, T2, FLAIR, GRE, and DWI.

Feasibility was qualitatively defined as detectable gadolinium enhancement signal into the sonicated targets on T1-weighted imaging immediately post-procedure, and resolution of the enhancement by the next day (reversibility criterion). This was further quantified by the gadolinium signal intensity ratio (IR) within a region of interest (ROI) in the targeted area, normalized to the contralateral, unsonicated

mirrored ROI. Square ROIs were created the size of and centred on the sonication targets, which are illustrated in Fig. S1. Exploratory measures (modified Ashworth Scale for spasticity, MoCA, ALSFRS-R, and EEG) were collected at baseline and at day 30. The final study event was a remote visit 60 days after the procedure to collect AEs and ALSFRS-R. In-person and remotely collected ALSFRS-R scores have excellent and comparable inter- and intra-rater reliability[54].

**Data analysis**. Aggregate data was reported as median and range for continuous variables and percentages for categorical variables. MR enhancement from gadolinium leakage post-sonication was used as a surrogate marker of BBB permeability, and this was quantified by the signal IR longitudinally. Task fMRIs were analysed using Statistical Parametric Model (SPM) version 12. Briefly, preprocessing consisted of motion correct, slice-time correction, functional realignment, co-registration to anatomic image, structural segmentation, functional and structural normalization, and spatial smoothing with 6 mm Gaussian kernel. The activation measure of interest was contrast of movement greater than fixation. The individual statistical parametric maps were generated from t-statistics after family wise error corrected $p < 0.05$, which was then overlaid on the patient specific structural scans to identify the gyrus activated by the motor task.

In order to assess the precentral gyrus cortical volume over time, we processed the structural T1-weighted MRIs with the longitudinal stream in Freesurfer surface-based analysis[55], which is documented and available online (http://surfer.nmr.mgh.harvard.edu/). The longitudinal stream increases the reliability of within-subject estimates by creating unbiased within-subject template space and image on which processing steps (e.g. skull stripping, atlas transforms and registration, and parcellations) are then performed.

**Reporting summary**. Further information on research design is available in the Nature Research Reporting Summary linked to this article.

## Data availability
The authors declare that all the data supporting the findings of this study are available within the paper and its supplementary information files. Study protocol, de-identified individual-participant data and data dictionary are available from the corresponding author upon request.

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

## Acknowledgements

This investigator-initiated study was funded by an ALS Society of Canada peer-reviewed grant and generosity of philanthropic gifts to the Sunnybrook Foundation by the Temerty, Stock, Lechem and Harquail families. The authors are grateful to InSightec® Ltd (Israel) for the regulatory and technical sponsorship throughout the study. Both InSightec® and the ALS Society of Canada had no role in study design, data collection, analysis, interpretation or reporting. The authors also would like to thank Ruby Endre and Garry Detzler for their technical support in the study. N.L. gratefully acknowledge the philanthropic support provided through the Sunnybrook Foundation and the Harquail Centre for Neuromodulation. Finally, we are grateful to the patients and their families for their involvement and contributions.

## Author contributions

A.A., K.H., T.M., S.E.B., I.A., N.L., and L.Z. contributed to the study conceptualization and design. A.A., Y.M., C.Hamani, M.L., C.Heyn, N.L., and L.Z. participated in data collection. Y.H. and K.H. provided technical focused ultrasound expertise. A.A., Y.M., N.L., and L.Z. had full access to the data and contributed to the analysis and interpretation. A.A. and Y.M. served as co-first authors; N.L. and L.Z. served as co-senior authors of the paper. All authors critically revised the paper.

## Competing interests

N.L., K.H., and S.E.B. have received an honorarium from the Focused Ultrasound Foundation for serving on an expert steering committee on focused ultrasound in Alzheimer's Disease. K.H. is an inventor on intellectual property owned by Brigham and Women's hospital in Boston and Sunnybrook Research Institute in Toronto related to intracranial focused ultrasound technology. A.A., Y.M., M.L., C.Hamani, Y.H., T.M., I.A., C.Heyn, N.L., and L.Z. declare no competing interests.
