## [Peer Review File · Nature Communications]

Reviewers' comments:

Reviewer #1 (Remarks to the Author):

This article demonstrates that it is possible to target ultrasound to a specific region of the cortex and opens the way for focused ultrasound-based interventions in neurological diseases such as degenerative conditions. ALS is a good potential candidate for this technology.

Preclinical data :

- Lines 267-268: unpublished data. Are those your data? Should note "Author's name et al., unpublished data" or "our unpublished data". The authors should also note the reasons for not publishing and a summary of the data.
- Preclinical data are of importance to justify the rationale of human transfer of BBB opening in ALS

ALS disease:

- Line 106 : "The driving influence of cortical motor neurons in ALS pathophysiology has also been demonstrated in human studies using transcranial magnetic stimulation, neuroimaging and electrophysiological measures". Upper motor neuron degeneration in ALS is indeed responsible for symptoms such as spasticity or pseudo-bulbar syndrome. However, the rationale stated by the authors is much more ambitious since they consider that the whole degenerative process in ALS is driven by upper motor neuron degeneration: This is far from being justified by the literature. Whether upper or lower motor neuron degenerates first remains a matter of debate. There is indeed a "driving forward" hypothesis , meaning that degeneration could start in the upper motor neuron and then "diffuse" to the lower motor neuron (through completely unknown mechanisms..). However, there is also a lot of data suggesting a "dying-back" mechanism, in particular suggesting that the lower motor neuron and/or the neuromuscular junction abnormalities are the very first events. Not targeting lower motor neuron degeneration, which is the main cause of functional impairment and death through respiratory insufficiency, is a limitation of the proposed approach. A putative effect on the lower motor neuron can be discussed but only in a putative manner.
- Line 281 : "only patients with spinal-onset ALS" appears in contradiction with the "driving forward" rationale of the study. If the goal was to assess safety on the cortical side of the disease, why wouldn't authors use spasticity patients in the study ? "for safety concerns" which makes perfect sense, but a little more explanation is needed considering the non-ALS specialized targeted readers of this journal. Authors specify that motor cortex atrophy was present, but there is no quantification of this effect in the article. This would support the safety of BBB opening in ALS.

- Lines 324-325: authors need to specify if the patients that were under treatment still when they did the BBB opening, with PK expectation blood level, and how long before the BBB opening was the treatments taken, including injection of Edaravone done in patient 3

Safety

- To my knowledge, I think that previous data on animals and humans do exist supporting that BBB opening in eloquent cortex is well tolerated. A reference would reinforce this message.
- Have pre versus post sonication Evoked Motor Potentials been performed to monitor safety? It would have made sense, especially since precise targeting was defined by an additional fMRI exam and since it is a regularly used in exams to monitor such patients.
- Lines 171-174: provide additional details on the occurrence of FLAIR signal. Did it take 3 weeks to resolve? Were any other medications given to patient (steroids?)
- Line 279 : “4 patients... typical number for first in human surgical trials”. This statement could be deleted.
- To envision efficacy of the technique in the future, wouldn't BBB opening sessions be repeated (as in Alzheimer's disease) ? A comment on this possibility should appear in the Discussion section.
- For this technique to be effective in the future, wouldn't BBB opening volumes be much larger than 0.35 cm³, which is far less than the solely hand or foot area volume (around 3cm³)? Please address. Additionally, is transcranial HIFU able to achieve this in the future?

Ultrasound

- Lines 160-161: The authors report a power output for the Insightec system of 8.0W. The authors should report a pressure value in MPa along with the ultrasound frequency here so that the results can be easily transposed/understood in the context of other ultrasound systems for the reader. A statement comparing these power levels to those used in other Insightec trials would also be helpful.
- Lines 166-168 : mean time duration of 66 min is for 1 target or for the four targets? It would be helpful to present the total duration of the whole patient's procedure starting from shaving to stereotactic frame removal.
- Line 216 : ALS is a cortical disease (grey). Would it be possible to display an ultrasound deposition map to see if BBB opening was effectively in the grey cortical matter and not only in the white matter? Especially regarding the line 368 “ in the middle of the gyrus” signaling more a white matter targeting, as Gd enhancements is in the figures. In line 369 : “brain regions containing sulci and vessels were avoided to mitigate the potential risk of hemorrhage” but the grey matter part of the cortex is always in contact with sulci and vessels. Can you provide rational for this statement?

Additionally, in the future, could BBB opening be performed only in sulci grey matter or also on top of gyri ?

- Lines 362-363: Please list what was drugs and doses were used for patient' comfort and for how long (analgesia and/or sedation drug for the procedure).
- Line 373: power ramp test to determine the optimal ultrasound power required for safe BBB opening. Can you provide additional details of this for the reader?
- Line 374: 4 point at the corners = 4 sonications procedures performed in the same time or separately ?

MRI

- Figures 2-4: IR increase of contrast enhancement appears around 10% and is hardly visible on patients 2 and 3. No statistical analysis or confident intervals are presented. Is a 10% enhancement significant? From what I have read on usBBB opening, the target of 20% increase is generally presented. Please provide additional details, especially if the goal of the technique is to have drug delivery in the future. A display of the US target would be useful. Also provide the size and shape of ROI on the images in Fig. 2 that are used for calculations shown in Fig. 4. The images are very difficult to interpret with the current contrast settings.
- Fig 4: "increased IRs were seen immediately after BBB opening"... "and returned to baseline within 24h". Apparently, subject 2 does not go back to baseline and some go lower than baseline, especially for subject 4. Why do 3/4 of the patients go lower than baseline after 30 days? The problem with this figure is that we do not know the "sensitivity" If they are all supposed to go to baseline then, subject 2 either did not go back to baseline at day 1 or did not have an increase at "immediate". Difficult to say. What difference is supposed to be significant? Also, indicate size of ROI used on Figs. 2 and 3. Also, a voxel by voxel comparison with a level of "maximum" gadolinium enhancement and not simply a comparison of ROI means would be helpful (or statement in the text).
- Line 227: "without radiological evidence of inflammation" needs more evidence since it is stated in line 228-230: "subject 4 had a FLAIR hyperintensity". The same patient in which had stronger visible Gd uptake. It is mentioned that this patient had a "comorbidity" (I assume this means in general, not after the procedure). How would comorbidities such as chronic pain would explain the hyper FLAIR signal observed?
- Lines 238-239: how can real-time MRI thermography and acoustic spectrum monitoring be performed in the same MRI time during the 90 sec BBB opening sonication? Does real-time monitoring during sonication include ultrasound monitoring and patient feedback but not MRI "tissue integrity"? Can tissue integrity be analyzed only after sonication finishes? Line 239-240 should segment the different monitoring technique according to sonication. A diagram of MRI acquisition duration also displaying the sonication period would be helpful for better understanding the techniques used.

Discussion

- Lines 286-287: The authors should also include clinical trials that are open with devices other than simply the Insightec systems in this list of NCT trials.

Reviewer #2 (Remarks to the Author):

This prospective, open-label, single-arm, single-centre first-in-human study is designed to evaluate the technical feasibility and safety of BBB opening in the primary motor cortex of subjects with ALS.

I certainly think that the relevance of this study is in the fact that it shows that BBB can be precisely, safely, and temporarily opened in a region in the primary motor cortex in ALS subjects using transcranial MRgFUS.

It is also the first use of MRgFUS to safely permeabilize fMRI-directed eloquent cortex in humans as prior studies targeted sub-cortical white matter regions.

It shows that the procedure is well tolerated with no SAEs.

Targeting of the motor cortex is achieved with tissue sonication which is well tolerated and no detectable hemorrhage or persistent lesions on longitudinal imaging are reported.

The Authors report previous published data on the same approach used in animal models for the delivery of treatments including antibodies which appear to have had an effect in prolonging survival, enhancing targeted brain bioavailability of large drugs.

I think this is an interesting, well-written and conducted study which deserve consideration for publication. The work is novel and a good contribution to a pioneering methodology which may have a profound impact in the treatment of neurodegenerative disorders.

However, I would also suggest that the manuscript is reviewed to include further discussion of different aspects and clarification about the set up of the neurological monitoring after procedure. In particular, I would suggest some more thoughts about the following points:

The Authors should discuss a bit more extensively about the merit of using their US-based procedure compared to other means of BBB permeabilization, including the use of pharmacological agents. The advantage of obtaining a focal effect in increased BBB permeability using MRgFUS should be discussed in light of the extension of neuropathology of ALS, which is known to affect upper and lower motor neurons and other cortical areas particularly in its clinical overlap with FTD.

There should be discussion regarding the most effective ways to look at potential injury at the site of treatment, beside the MR based measures (IR) and the clinical assessment to test features of upper motor neuron involvement following treatment. For example, would the use of DTI add anything to this assessment? The Authors provide mostly measures of clinical evaluation of global neurological impairment used in the functional rating of ALS (ALSFRSR), upper motor neuron function (spasticity scale) and cognitive function (MoCA) pre and post-treatment. These measures can lack sensitivity and are also examiner-dependent.

The Authors refer to previous published data about AAV9 introduction via lumbosacral intrathecal administration which nicely diffuse across multiple anterior horn levels of the spinal cord but has little diffusion into the motor cortex. This observation is clearly in favour of using MRgFUS for a more localised effect on BBB but would also raise the question to whether MRgFUS would work well in different neuroanatomic structures like in the spinal cord. Considering that lower motor neuron involvement remains a substantial aspect of the disease pathogenesis, what are the Author's thoughts about the feasibility to use this technique for spinal cord BB modification?

Also ALS is notoriously a non-cell autonomous disease although, admittedly, the end point of the pathological process and the ultimate source of irreversible neurological dysfunction is motor cell loss. We know little about the origin of the pathological process and glia as well as axons may be deeply located. What is in the Authors opinion (also based on animal models studies and previous published data) the depth of tissue/BBB involvement which may condition the degree of penetration into cortical and sub-cortical tissue of a therapeutic agent? They refer in their manuscript to the aim of opening a maximum BBB volume of 1 cm³.

The statement "...the lack of gadolinium enhancement in the brain and spinal cord of ALS subjects indicates a clinically impermeable BBB in ALS, which is in contrast to inflammatory CNS diseases such as multiple sclerosis (MS) which are characterized by gadolinium enhancement...". It would be important also to note that there are numerous reports suggesting a BBB damage in ALS and that

this has been identified using different means of investigation which are not necessarily dependent on gadolinium enhancement. Therefore, it is difficult to be absolutely sure about the degree of BBB permeability in a pathological condition like ALS.

The Authors emphasize the theory of an upper motor neuron dominant origin of ALS, which would justify in full the use of their technique for transitory permeabilization of the BBB. I think it may be worthwhile also mention the theory of distal axonopathy, which includes the concept of neuromuscular junction primary involvement and a caudal to rostral progression of the disease

I would also like to refer to the statement “..Longitudinal neurological assessments revealed no significant change in disease progression as indicated by ALSFRS-R and MRC scores for the limb contralateral to the sonicated cortex...”

I think the Authors may want to say that in their view, there was no procedure-induced acceleration of disease progression, as the disease would be expected to progress over time (we would not know whether BBB permeabilization in itself has a positive disease modifying effect). I think it may be worth specifying what is the length of longitudinal follow-up over time they are referring to and state what parameter to evaluate disease progression they have used. If this is ALSFRSR, then what may be useful is to plot the ALSFRSR temporal slope pre and post treatment to show the relative steady progression of the disease or the lack of progression in the pre to post-procedure follow-up. The Authors have accurately selected their ALS subjects, with a range of disease durations to time of treatment of 4 to 4.5 years (with an average ALSFRSR at treatment of 30 to 35), which is likely to be a stage 2 or 3 of the disease, and modified Ashworth score of 0 at the time of treatment (to allow for a better characterization of potential cortico-spinal / upper motor neuron changes after treatment).

Andrea Malaspina

Reviewer #1 (Remarks to the Author):

This article demonstrates that it is possible to target ultrasound to a specific region of the cortex and opens the way for focused ultrasound-based interventions in neurological diseases such as degenerative conditions. ALS is a good potential candidate for this technology.

[We thank the reviewer for acknowledging the potential application of the MRgFUS platform for therapeutic delivery in neurodegenerative diseases, particularly ALS. Also, thank you for your time and thoughtful comments which has improved our manuscript. We have carefully reviewed your suggestions and substantial revisions have been made to the revised manuscript in response.]

Preclinical data :

- Lines 267-268: unpublished data. Are those your data? Should note "Author's name et al., unpublished data" or "our unpublished data". The authors should also note the reasons for not publishing and a summary of the data.

[Thank you for highlighting this point. The first successful and safe preclinical application of MRgFUS in the spinal cord of wild rats was led by Dr. Aubert, a co-author on this manuscript (Weber-Adrian 2015). This was later replicated by Payne et al., 2017 and these references were added.

Dr. Erwin, MW and collaborators from University of Toronto presented the preliminary feasibility and safety of blood-spinal cord barrier opening using MRgFUS in a small sample of ALS SOD1 mice at the 12th Annual ALS Canada Research Forum, Toronto, Canada on May 1, 2016 (presentation title: "Non-surgical image guided transplantation of stem cells: thinking outside the box for a revolution in the treatment of ALS"). We do not have a summary of the data. Dr. Erwin has left the University of Toronto and the study has not been published.

We changed the manuscript as follows and included the two published studies:

286-287: MRgFUS blood-spinal cord barrier opening has not yet been developed in humans, but has been performed safely in the spinal cord of non-ALS rodent models^{44,45}.

New references:

44. Weber-Adrian, D. et al. Gene delivery to the spinal cord using MRI-guided focused ultrasound. *Gene Ther.* **22**, 568–577 (2015).

45. Payne, A. H. et al. Magnetic resonance imaging-guided focused ultrasound to increase localized blood-spinal cord barrier permeability. *Neural Regen. Res.* **12**, 2045–2049 (2017).

- Preclinical data are of importance to justify the rationale of human transfer of BBB opening in ALS

[We agree with the importance of preclinical studies to justify translational clinical studies.

We formulated the rationale for our trial in the introduction and discussion as follows:

- 1. We presented a number of preclinical studies of potential candidates of ALS disease-modifying biotherapeutics that required invasive surgery for targeted motor cortex delivery;*
- 2. We introduced the MRgFUS technology and the preclinical data establishing its application as a targeted delivery platform for multiple therapeutic strategies, such as antibodies, gene therapy and stem cells. We also cited the first human experience using MRgFUS BBB opening in non-eloquent brain regions. Our co-authors, Drs. Huang, Aubert, Meng, Lipsman, Black and Hynynen pioneered most of these preclinical and clinical developments;*
- 3. We hypothesized that the MRgFUS BBB opening would be safe in the motor cortex of patients with ALS in this novel pilot trial, in preparation for a larger trial of MRgFUS coupled with a therapeutic.*
- 4. Additional preclinical studies on ALS SOD1 mice models are planned following the publication of this manuscript to inform the most promising therapeutic to be coupled with MRgFUS brain delivery.]*

ALS disease:

- Line 106 : “The driving influence of cortical motor neurons in ALS pathophysiology has also been demonstrated in human studies using transcranial magnetic stimulation, neuroimaging and electrophysiological measures”.

Upper motor neuron degeneration in ALS is indeed responsible for symptoms such as spasticity or pseudo-bulbar syndrome. However, the rationale stated by the authors is much more ambitious since they consider that the whole degenerative process in ALS is driven by upper motor neuron degeneration: This is far from being justified by the literature. Whether upper or lower motor neuron degenerates first remains a matter of debate. There is indeed a “driving forward” hypothesis, meaning that degeneration could start in the upper motor neuron and then “diffuse” to the lower motor neuron (through completely unknown mechanisms..). However, there is also a lot of data suggesting a “dying-back” mechanism, in particular suggesting that the lower motor neuron and/or

the neuromuscular junction abnormalities are the very first events. Not targeting lower motor neuron degeneration, which is the main cause of functional impairment and death through respiratory insufficiency, is a limitation of the proposed approach. A putative effect on the lower motor neuron can be discussed but only in a putative manner.

[Thank you for highlighting the ongoing debate between the dying-forward and dying-back hypotheses in the ALS pathology. We changed the introduction in the revised manuscript to acknowledge this uncertainty as follows:

Lines 99-107: "While preclinical studies in ALS rodent models have primarily focused on the distal-to-proximal motor neuron dying-back hypothesis¹, recent data support a key pathological role of UMN in the motor cortex.² Signs of cortical neuronal and glial dysfunction have been observed early in the disease course and preceded spinal cord neuronal degeneration in ALS rodent models^{3,4}, supporting a motor cortex driven dying-forward hypothesis.² Early cortical motor neuron dysfunction in ALS has also been demonstrated in human studies using transcranial magnetic stimulation, neuroimaging and electrophysiological measures.^{2,5,6} Therefore, interventions that target cortical motor neurons alone or in combination with LMN-directed therapies may be essential in slowing or halting disease progression."

While upper motor neuron (UMN) degeneration is a key pathological feature of ALS, even in patients with clinical signs of pure lower motor neuron (LMN) signs (Neurology. 2003 Apr 22;60(8):1252-8, PMID 12707426), we agree that therapeutic approaches targeting both UMN and LMN might have a synergistic effect. We included this hypothesis as a new paragraph in the revised discussion as follows:

Lines 283-291: "To enhance a potential synergistic effect on UMNs and LMNs, targeted brain therapeutic delivery using MRgFUS can be coupled with non-targeted, lumbosacral intrathecal therapeutic injections, which have unreliable motor cortex penetration and are currently being used in ALS clinical trials. MRgFUS blood-spinal cord barrier opening has not yet been developed in humans, but has been trialled safely in the spinal cord of non-ALS rodent models^{41,42}. An early study demonstrated enhanced chemotherapy delivery using MRgFUS blood-spinal cord barrier opening in rodents.⁴³ However, the bony vertebrae of the spinal column in primates present a technical challenge for focusing the ultrasound and further validation and optimization studies are necessary before translation to humans.⁴⁴ "

New References:

42. Weber-Adrian, D. et al. Gene delivery to the spinal cord using MRI-guided focused ultrasound. *Gene Ther.* **22**, 568–577 (2015).
43. Payne, A. H. et al. Magnetic resonance imaging-guided focused ultrasound to increase localized blood-spinal cord barrier permeability. *Neural Regen. Res.* **12**, 2045–2049 (2017).
44. O'Reilly, M. A. et al. Preliminary Investigation of Focused Ultrasound-Facilitated Drug Delivery for the Treatment of Leptomeningeal Metastases. *Sci. Rep.* **8**, 9013 (2018).
45. S. P. Fletcher & M. A. O'Reilly. Analysis of Multifrequency and Phase Keying Strategies for Focusing Ultrasound to the Human Vertebral Canal. *IEEE Trans. Ultrason. Ferroelectr. Freq. Control* **65**, 2322–2331 (2018).

- Line 281 : “only patients with spinal-onset ALS” appears in contradiction with the “driving forward” rationale of the study. If the goal was to assess safety on the cortical side of the disease, why wouldn't authors use spasticity patients in the study ? “for safety concerns” which makes perfect sense, but a little more explanation is needed considering the non-ALS specialized targeted readers of this journal. Authors specify that motor cortex atrophy was present, but there is no quantification of this effect in the article. This would support the safety of BBB opening in ALS.

[Thank you for this comment and we agree that the term “spinal-onset ALS” may confuse the non-ALS specialized readers. For clarity, we changed “spinal-onset” to another well-accepted terminology, “limb-onset ALS”. Both terms refer to the site of first reported weakness caused either by upper or lower motor neuron dysfunction (i.e. caused by brain and/or spinal cord motor neuron degeneration) .

Cortical atrophy was determined qualitatively by the study radiologist. There was no longitudinal change in the radiological appearance of the sonicated primary motor cortex region.

Revised discussion:

Lines 295-296: “For safety concerns, only patients with limb-onset ALS and severe weakness were included and only a small volume of the primary motor cortex was targeted.”

We limited the inclusion criterion to participants with a diagnosis of laboratory-supported probable, clinically probable or definite ALS. According to the revised El Escorial criteria,

these categories require clinical evidence of upper motor neuron dysfunction in at least one body region. Pure lower motor neuron phenotype of ALS does not fall in these diagnostic categories. We also clarified this point in the revised Result and Method sections:

Results:

Lines 164-166: “Four eligible subjects (two women and two men) diagnosed with ALS (probable El Escorial criteria) and clinical evidence of upper motor neuron dysfunction were enrolled in the study (Figure 1).”

Methods:

Lines 346-349: “Each was diagnosed with laboratory-supported probable, clinically probable or definite ALS according to the revised El Escorial criteria⁵¹, which requires evidence of UMN dysfunction in at least one body region.”

Lastly, we would like to thank the reviewer for suggesting cortical volume analyses as a safety measure to enhance our study. The baseline primary motor cortex atrophy was diagnosed by the study neuroradiologist (Dr. Heyn) and the longitudinal cortical volumes were analysed and estimated using Freesurfer pipeline. In this additional analysis in the revision, longitudinal measurements did not show any significant change in primary motor cortex volume. This result is presented as a new figure S3.

Lines 170-171: “Cortical atrophy of the hand functional region was also noted on visual inspection at baseline in subject three, who presented with arm-onset phenotype.”

Line 197-199: “In all participants, follow-up MR sequences did not demonstrate parenchymal or subarachnoid hemorrhage, ischemia, gliosis or worsened precentral gyrus cortical atrophy (Figure S3) up to 30 days following the intervention.”

Line 562-568: In order to assess the precentral gyrus cortical volume over time, we processed the structural T1-weighted MRIs with the longitudinal stream in Freesurfer, which is documented and available online (<http://surfer.nmr.mgh.harvard.edu/>). The details of the Freesurfer surface-based analysis pipeline have been described previously⁵⁵. The longitudinal stream increases the reliability of within-subject estimates by creating unbiased within-subject template space and image on which processing steps (e.g. skull stripping, atlas transforms and registration, and parcellations) are then performed.]

- Lines 324-325: authors need to specify if the patients that were under treatment still when they did the BBB opening, with PK expectation blood level, and how long before the BBB opening was the treatments taken, including injection of Edaravone done in patient 3

[Thank you for this important comment. Although MRgFUS BBB opening can increase the bioavailability of riluzole and edaravone, these drugs were not administered on the day of the MRgFUS procedure to minimize potential confounders, given the primary trial outcome was to determine the safety of the device and BBB opening alone. Future single or repeated MRgFUS BBB opening trials in ALS should couple the infusion of these medications 60 minutes before BBB opening as indicated by oral riluzole peak dose and riluzole and edaravone half-lives. We clarified this point in the revised manuscript:

Lines 352-356: "Participants on a stable dose of oral riluzole (up to 50mg twice daily) for at least 30 days or intravenous edaravone for at least two cycles were eligible for the study. Edaravone and were not administered on the day of the MRgFUS procedure and restarted after discharge."

Safety

- To my knowledge, I think that previous data on animals and humans do exist supporting that BBB opening in eloquent cortex is well tolerated. A reference would reinforce this message.

[We agree that eloquent cortical and subcortical regions, e.g. hippocampus, striatum, brainstem, have been previously targeted in MRgFUS animal studies. We have added this information to the introduction as follows:

Introduction:

Lines 136-138: "This has been shown to enhance drug delivery to multiple brain regions in small-to-large animals, including cortical targets¹⁹, the hippocampus²⁰, striatum^{20,21}, and brainstem²²."

In humans, Carpentier et al. 2016 and 2019 described BBB opening in language and motor areas using an invasively-implanted pulsed (not focused) ultrasound. Also, they have not localized the target via functional methods, such as fMRI. Therefore, BBB opening from pulsed ultrasound is less controlled and spatially selective as compared to MRgFUS. Although these data supported the safety of microvascular disruption in eloquent brain regions, our study represented the first human application of a non-invasive MRgFUS

technique targeting the primary motor cortex, guided by fMRI. These differences were explained in the revised manuscript as follows:

Introduction:

Lines 138-156: "Although safe BBB opening in eloquent and non-eloquent brain regions has been demonstrated in subjects with high-grade gliomas using surgically implanted intracranial pulsed ultrasound device^{24,25}, transcranial MRgFUS BBB opening is a non-invasive approach and has optimal spatial resolution for discrete and precise BBB targeting in eloquent regions, such as the primary motor cortex. In recent pilot clinical trials, non-invasive MRgFUS BBB opening in small and non-eloquent regions in and around gliomas²⁶ and in the frontal lobe of patients with Alzheimer's disease (AD)¹⁴ was safe and transient."

We restated the novelty of our study as follows:

Abstract

Lines 80-83: Here, we conducted a single-arm, first-in-human trial to investigate the safety and feasibility of MRgFUS-induced BBB opening in eloquent primary motor cortex in four volunteers with amyotrophic lateral sclerosis (ALS). We also evaluated the first attempt to use functional MR mapping for precise targeting of the motor cortex in subjects ALS.

Introduction:

Lines 159-162: This study represents the first trial testing the feasibility and safety of transient permeabilization of the BBB using MRgFUS in subjects with ALS. It is also the first attempt to apply this novel technology to eloquent cortical and subcortical tissue in human subjects as primary motor cortex was targeted using functional MRI (fMRI).

Discussion:

Lines 216-218: "It also represents the first clinical application of non-invasive, transcranial MRgFUS to safely permeabilize fMRI-directed eloquent brain regions as prior MRgFUS studies targeted sub-cortical white matter regions in humans.^{13,25"}

New references:

20. Jordão, J. F. et al. Antibodies targeted to the brain with image-guided focused ultrasound reduces amyloid-beta plaque load in the TgCRND8 mouse model of Alzheimer's disease. PLoS ONE 5, e10549 (2010).

21. Thévenot, E. et al. Targeted delivery of self-complementary adeno-associated virus serotype 9 to the brain, using magnetic resonance imaging-guided focused ultrasound. *Hum Gene Ther* **23**, 1144–1155 (2012).
22. Noroozian, Z. et al. MRI-Guided Focused Ultrasound for Targeted Delivery of rAAV to the Brain. *Methods Mol. Biol. Clifton NJ* **1950**, 177–197 (2019).
23. Alli, S. et al. Brainstem blood brain barrier disruption using focused ultrasound: A demonstration of feasibility and enhanced doxorubicin delivery. *J Control Release* **281**, 29–41 (2018).
24. Carpentier, A. et al. Clinical trial of blood-brain barrier disruption by pulsed ultrasound. *Sci. Transl. Med.* **8**, 343re2 (2016).
25. Idbaih, A. et al. Safety and Feasibility of Repeated and Transient Blood–Brain Barrier Disruption by Pulsed Ultrasound in Patients with Recurrent Glioblastoma. *Clin. Cancer Res.* (2019). doi:10.1158/1078-0432.CCR-18-3643]

- Have pre versus post sonication Evoked Motor Potentials been performed to monitor safety? It would have made sense, especially since precise targeting was defined by an additional fMRI exam and since it is a regularly used in exams to monitor such patients.

[Evoked motor potentials were not performed in this pilot trial. Instead, we included pre and post-sonication EEG testing to assess for BBB-opening-induced epileptiform discharges. In our next MRgFUS BBB opening trial coupled with the injection of a therapeutic in patients with ALS, we plan to include threshold-tracking transcranial magnetic stimulation measures of upper motor neuron function, such as motor evoked potential amplitude, central motor conduction intervals, and short-interval intracranial inhibition (SICI). The latter is a measure of cortical excitability and GABA activity (Vucic S, Kiernan MC. Novel threshold tracking techniques suggest that cortical hyperexcitability is an early feature of motor neuron disease. Brain. 2006 Sep;129(Pt 9):2436-46. Epub 2006 Jul 10.)

Revised discussion:

Lines 325-328: “Neuroinflammation and endothelial, neuronal and glial function should be further investigated in upcoming MRgFUS trials using advanced MR imaging and spectroscopy, cortical excitability measures (e.g., short intracortical interval parameter of threshold-tracking transcranial magnetic stimulation), PET imaging (e.g. TPSO⁵¹), along with serum and cerebrospinal fluid biomarkers.”]

- Lines 171-174: provide additional details on the occurrence of FLAIR signal. Did it take 3 weeks to resolve? Were any other medications given to patient (steroids?)

[The punctate parenchymal FLAIR hyperintensity was noted on day-1 and day-7 MR scans (supplementary information, Figure S2), corresponding to a partial volume of the sonicated region. This finding has resolved on the day-30 scan without any specific treatment. We provided additional details in the revised manuscript as follows:

Results section:

Lines 191-197: “In subject four, a transient and asymptomatic hyperintense sulcal and parenchymal fluid-attenuated inversion recovery (FLAIR) signal was observed in a portion of the sonicated region on the day 1 and resolving on day 7 MR scans, without any accompanying changes on T2, T2 gradient echo (GRE), or diffusion weight imaging (DWI) sequences (Figure S2). This radiologic finding did not correlate with any new symptoms, neurological signs, or focal EEG changes and completely resolved at day-30 MRI.”*

Discussion:

Lines 234-239: “One participant (subject four) developed a delayed and self-limited, small parenchymal FLAIR hyperintensity within the sonicated region. This asymptomatic finding did not require any intervention and was inconsistent with hemorrhage, ischemia, or gliosis and completely resolved on imaging obtained 30 days following the intervention. While the aetiology and clinical significance is uncertain, the finding may represent serum extravasation of gadolinium, albumin and/or immunoglobulins from BBB opening.^{27,28”]}

- Line 279 : “4 patients... typical number for first in human surgical trials”. This statement could be deleted.

[This has been deleted.]

- To envision efficacy of the technique in the future, wouldn't BBB opening sessions be repeated (as in Alzheimer's disease) ? A comment on this possibility should appear in the Discussion section.

[MRgFUS procedures will have to follow the dosing protocol of the therapeutic, which in many cases, will be repetitive. This was added to the revised discussion as follows:

Lines 253-255: “Multiple MRgFUS procedures are also feasible for therapeutics with repeated dosing as demonstrated in a pilot MRgFUS trial in subjects with AD¹³ and preclinical studies^{35,36}”

New references:

38. Alkins, R., Burgess, A., Kerbel, R., Wels, W. S. & Hynynen, K. Early treatment of HER2-amplified brain tumors with targeted NK-92 cells and focused ultrasound improves survival. *Neuro-Oncol.* **18**, 974–981 (2016).

39. Nisbet, R. M. et al. Combined effects of scanning ultrasound and a tau-specific single chain antibody in a tau transgenic mouse model. *Brain J. Neurol.* **140**, 1220–1230 (2017).]

- For this technique to be effective in the future, wouldn't BBB opening volumes be much larger than 0.35 cm³, which is far less than the solely hand or foot area volume (around 3cm³)? Please address. Additionally, is transcranial HIFU able to achieve this in the future?

[Thank you for this suggestion. Animal studies showed whole hemispheric BBB volumes can be targeted with MRgFUS. Also, BBB opening of larger brain volumes, through multiple sonications and larger spacing, is an objective of currently ongoing clinical studies. This is discussed in the revised manuscript as follows:

Lines 296-306: “While the minimal volume of BBB opening for sufficient therapeutic distribution is unknown and likely depends on drug properties, future trials will explore the safety and feasibility of sonicating larger volumes or the entirety of the primary motor cortex, along with frontotemporal regions that are also affected in ALS with frontotemporal dementia. Current preclinical evidence support the feasibility of targeting large hemispheric volumes in canines, whose brain dimensions more closely resemble that of humans.⁴⁶ Further, the clinical MRgFUS device can achieve a larger volume of BBB opening through a summation of targets and wider spacing of spots within each target.¹⁴ Ongoing clinical trials using focused ultrasound (NCT03616860, NCT03712293, NCT03714243, NCT03626896) as well as pulsed ultrasound (NCT03744026) are underway to treat larger tumour regions and peri-tumour tissue to enhance chemotherapy delivery.”

New reference:

46. O'Reilly, M. A. et al. Investigation of the Safety of Focused Ultrasound-Induced Blood-Brain Barrier Opening in a Natural Canine Model of Aging. *Theranostics* **7**, 3573–3584 (2017).]

Ultrasound

- Lines 160-161: The authors report a power output for the Insightec system of 8.0W. The authors should report a pressure value in MPa along with the ultrasound frequency here so that the results can be easily transposed/understood in the context of other ultrasound systems for the reader. A statement comparing these power levels to those used in other Insightec trials would also be helpful.

[Thank you for this suggestion. Ex vivo measurements are available from previous studies by our group. However, in vivo pressures will differ depending a number of factors including individual skull variabilities, target location, and exact transducer placement. Therefore, for reproducibility and replication by other groups, we recommend the procedure by acoustic feedback. We provide the following detail in our methods:

Line 418-424: For transcranial focused ultrasound applications, accurate estimation of in vivo tissue pressure for individual subjects at this level of power is limited given highly variable skull geometry, density, and target locations. Therefore, the optimal power for BBB opening was determined by cavitation feedback at each target through a ramp test.^{27,14} This involves applying short sonications with 5% power increments until the device's hydrophones detect sub-harmonic acoustic feedback from the target, indicating the cavitation threshold. The optimal power for BBB opening was estimated as 50% of this threshold."

Table S1. Ultrasound parameters used for MRgFUS induced opening of the BBB in four subjects. For reference, 5W of sonication power delivered by the clinical system at 220kHz through an ex vivo human skull translated to 500kPa measured by hydrophone at the center of the skull²⁷.]

- Lines 166-168 : mean time duration of 66 min is for 1 target or for the four targets? It would be helpful to present the total duration of the whole patient's procedure starting from shaving to stereotactic frame removal.

[We clarified this ambiguity in the revised manuscript as follows:

Lines 186-188: "Median duration in the MRI suite was 66 minutes (range: 43–173), which included acquisition of planning sequences, targeting and sonications. Median duration of pre- and post-sonication MR scanning was 115 minutes (range: 90–147)."]

- Line 216 : ALS is a cortical disease (grey). Would it be possible to display an ultrasound deposition map to see if BBB opening was effectively in the grey cortical matter and not only in the white matter? Especially regarding the line 368 “ in the middle of the gyrus” signaling more a white matter targeting, as Gd enhancements is in the figures. In line 369 : “brain regions containing sulci and vessels were avoided to mitigate the potential risk of hemorrhage” but the grey matter part of the cortex is always in contact with sulci and vessels. Can you provide rational for this statement? Additionally, in the future, could BBB opening be performed only in sulci grey matter or also on top of gyri ?

[Thank you for this suggestion. We now provide figures to show the ultrasound targets, from the InSightec system planning console (Figure S1). We agree that the targets cover a larger amount of white matter, but overall a mixture of white and grey matter is sonicated. Figure 2 and 3, in combination with the new figure S1, will help clarify the targeted cortical and subcortical regions. Furthermore, as explained in response to the first comment under “Ultrasound”, it is difficult to determine accurate in vivo pressure for transcranial application in human subjects.

Figure S1. Target placement for each subject overlapped onto intraprocedural axial T2-weighted MRIs. Each green box represents the target with 5 mm sides, centred over the arm or leg area as mapped by task fMRI.

By avoiding sulci and vessels, we mean that the target box avoided any sulci or visible vessels on MRI. We ensured through visual inspection that the box did not overlap any voxels containing sulci or visible vessels. This has been clarified in the revised manuscript:

Lines 399-403: “To achieve BBB opening within the arm or leg regions, two sonication targets of 5 x 5 x 7 mm (175 mm³ each, 350 mm³ total) were placed over the arm or leg region as mapped by task fMRI, avoiding overlap with any voxels containing sulci and visible vessels within two contiguous axial slices as to mitigate the potential risk of hemorrhage.”]

- Lines 362-363: Please list what was drugs and doses were used for patient’ comfort and for how long (analgesia and/or sedation drug for the procedure).

[Medications administered for patient’s comfort and AE management during the intervention are listed in Table S2 of the revised supplementary information.]

- Line 373: power ramp test to determine the optimal ultrasound power required for safe BBB opening. Can you provide additional details of this for the reader?

[The ramp test involves applying short sonications with 5% power increments until the device's hydrophones detect sub-harmonic acoustic feedback from the target, indicating cavitation. The optimal power for BBB opening is estimated to be 50% of this threshold as informed by preclinical studies. Details of ramp test were included in the revised Methods:

Lines 421-424: "This involves applying short ultrasound sonications with 5% power increments until the device's hydrophones detect sub-harmonic acoustic feedback from the target, indicating the cavitation threshold. The optimal power for BBB opening was estimated as 50% of this threshold."

- Line 374: 4 point at the corners = 4 sonications procedures performed in the same time or separately ?

[Each BBB opening volume, i.e. 5x5x7 mm, is composed of four points of sonication, which are targeted sequentially by the transducers during each 90s sonication. We have reworded previous explanation on page 21, which will hopefully clarify the ambiguity. A visualization of this protocol is also demonstrated by Figure S4.

Revised Methods:

Lines 403-404: "Within each sonication target, the focused ultrasound beam was electronically steered among four spots within the target as represented in Figure S4."

Figure S4. Schematic diagrams of MRgFUS target and sonications. A) Two 175 mm³ volumes or targets were placed over the primary arm or leg region mapped by task fMRI. Each target consists of four spots (marked) the device sonicates sequentially, switching every 300 ms, as described in B. Ultrasound was delivered with pulse repetition frequency of 0.3 Hz and 0.88% duty cycle per each of the four spots within a target. Each sonication cycle (total duration of 90s) was performed with an intravenous injection of 4 μ l/kg perflutren microbubbles (Definity®, Lantheus Medical Imaging, USA).]

MRI

- Figures 2-4: IR increase of contrast enhancement appears around 10% and is hardly visible on

patients 2 and 3. No statistical analysis or confident intervals are presented. Is a 10% enhancement significant? From what I have read on usBBB opening, the target of 20% increase is generally presented. Please provide additional details, especially if the goal of the technique is to have drug delivery in the future. A display of the US target would be useful. Also provide the size and shape of ROI on the images in Fig. 2 that are used for calculations shown in Fig. 4. The images are very difficult to interpret with the current contrast settings.

[Thank you for these suggestions. We have now adjusted the contrast of figures 2 and 3 to improve visibility.]

While it is true that 10% enhancement is lower than the one seen in preclinical studies, this is due to a combination of factors: 1) A delay of contrast T1 imaging with head coil relative to time of BBB opening is longer than what can be managed for animal studies. In the clinical setting, following the sonications, there is an interval for participant's rest and headframe removal before enhanced T1 acquisition with the head coil. ALS-related immobility prolonged this interval. 2) Given standardized dynamic-contrast enhanced imaging paradigm was not performed in this study, greater intra- and inter-individual variability in IR is expected. These are now discussed as limitations:

Lines 307-313: "The qualitative and semi-quantitative assessment of BBB permeability through contrast T1-weighted MRI is also a limitation in this study. Because the peak gadolinium enhancement is expected to be immediately after BBB opening, a delay between sonications and contrast MRI acquisition due to patient transfer, head frame removal, and rest will contribute to decreased visualized IR in the sonicated target. In future MRgFUS BBB opening trials, dynamic-contrast enhanced MRI sequence along with MR coil improvements can more quantitatively and accurately assess BBB permeability."

The ROIs were made to the size of the target (Figure S1), which was 5 by 5 mm square. Explanation of how ROIs were drawn for Figure 2 and 3 is added to the revised methods:

Line 536-537: "Square ROIs were created the size of and centred on the sonication targets, which are illustrated in Figure S1."]

- Fig 4: "increased IRs were seen immediately after BBB opening" ... "and returned to baseline within 24h". Apparently, subject 2 does not go back to baseline and some go lower than baseline, especially for subject 4. Why do 3/4 of the patients go lower than baseline after 30 days? The

problem with this figure is that we do not know the “sensitivity” If they are all supposed to go to baseline then, subject 2 either did not go back to baseline at day 1 or did not have an increase at “immediate”. Difficult to say. What difference is supposed to be significant? Also, indicate size of ROI used on Figs. 2 and 3. Also, a voxel by voxel comparison with a level of “maximum” gadolinium enhancement and not simply a comparison of ROI means would be helpful (or statement in the text).

[We appreciate your comments. These observations are also likely associated with intra-individual variability in the normalized IR and lack of a more quantitative dynamic-contrast enhanced imaging methodology in this pilot trial. We aimed to demonstrate the feasibility of BBB opening using enhanced T1 imaging in a qualitative (more than quantitative) manner (figure 2 and 3). Figure 4 plot is essentially descriptive aiming to facilitate the interpretation of longitudinal enhanced MR imaging. Therefore, we did not perform any significance testing in this small pilot trial to avoid type 1 errors and misleading inferences.

The ROI was defined above.]

- Line 227: “without radiological evidence of inflammation” needs more evidence since it is stated in line 228-230: “subject 4 had a FLAIR hyperintensity”. The same patient in which had stronger visible Gd uptake. It is mentioned that this patient had a “comorbidity” (I assume this means in general, not after the procedure). How would comorbidities such as chronic pain would explain the hyper FLAIR signal observed?

[Thank you very much for highlighting these points. We changed the wording in the revised manuscript in a conservative manner as follows:

Lines 224-234: “While there is preclinical and post-mortem evidence suggesting endothelial cell dysfunction and impaired BBB repair mechanisms in ALS⁷, in all participants of this study, BBB permeabilization was reversible, without radiological evidence of significant inflammation or tissue damage 30 days after the procedure.”

The listed comorbidities in Table 1 referred to pre-existing conditions, which were not influenced by the procedure. We changed this term to “Pre-existing Conditions” in the revised manuscript to avoid ambiguity.]

- Lines 238-239: how can real-time MRI thermography and acoustic spectrum monitoring be performed in the same MRI time during the 90 sec BBB opening sonication? Does real-time

monitoring during sonication include ultrasound monitoring and patient feedback but not MRI “tissue integrity”? Can tissue integrity be analyzed only after sonication finishes? Line 239-240 should segment the different monitoring technique according to sonication. A diagram of MRI acquisition duration also displaying the sonication period would be helpful for better understanding the techniques used.

[Real-time monitoring during sonications include acoustic monitoring, MR thermometry, and patient feedback. There is no MR interference to cavitation receivers built-in the FUS array, so MR thermometry can be performed during sonications while cavitation receivers were acquiring signals. MR thermometry used the standard FSPGR (fast spoiled gradient echo) sequence with TR = 26ms, TE = 13ms. This is clarified on line 399. Tissue integrity can be better assessed with T2 sequence immediately after sonications, page 22, line 402.]*

The sonication paradigm is now better represented by a diagram, added as panel B to Figure S4 in the supplementary information.]

Discussion

- Lines 286-287: The authors should also include clinical trials that are open with devices other than simply the Insightec systems in this list of NCT trials.

[Thank you for the suggestion. We have added two NCT trials using non-Insightec ultrasound device for BBB opening for chemotherapy delivery to the list (line 288). NCT03626896 employs the NaviFUS system, and NCT03744026 employs the Carthera system, an implanted ultrasound device, using pulsed, not focused ultrasound.]

Reviewer #2 (Remarks to the Author):

This prospective, open-label, single-arm, single-centre first-in-human study is designed to evaluate the technical feasibility and safety of BBB opening in the primary motor cortex of subjects with ALS.

I certainly think that the relevance of this study is in the fact that it shows that BBB can be

precisely, safely, and temporarily opened in a region in the primary motor cortex in ALS subjects using transcranial MRgFUS.

It is also the first use of MRgFUS to safely permeabilize fMRI-directed eloquent cortex in humans as prior studies targeted sub-cortical white matter regions.

It shows that the procedure is well tolerated with no SAEs.

Targeting of the motor cortex is achieved with tissue sonication which is well tolerated and no detectable hemorrhage or persistent lesions on longitudinal imaging are reported.

The Authors report previous published data on the same approach used in animal models for the delivery of treatments including antibodies which appear to have had an effect in prolonging survival, enhancing targeted brain bioavailability of large drugs.

I think this is an interesting, well-written and conducted study which deserve consideration for publication. The work is novel and a good contribution to a pioneering methodology which may have a profound impact in the treatment of neurodegenerative disorders.

[Many thanks for the very supportive comments and for highlighting the novelty and potential impact of our work.]

However, I would also suggest that the manuscript is reviewed to include further discussion of different aspects and clarification about the set up of the neurological monitoring after procedure. In particular, I would suggest some more thoughts about the following points:

The Authors should discuss a bit more extensively about the merit of using their US-based procedure compared to other means of BBB permeabilization, including the use of pharmacological agents. The advantage of obtaining a focal effect in increased BBB permeability using MRgFUS should be discussed in light of the extension of neuropathology of ALS, which is known to affect upper and lower motor neurons and other cortical areas particularly in its clinical overlap with FTD.

[Thank you for this suggestion. We have modified our Introduction to acknowledge non-targeted pharmacological BBB technologies, such as transferrin/insulin receptor conjugation, and compare them to MRgFUS:]

Lines 121-125: Alternatively, non-invasive delivery methods across the BBB via facilitated transcellular transport (e.g., drug modifications with transferrin or insulin receptors¹²), drug efflux inhibition (e.g., elacridar⁹), or diffuse BBB breakdown (e.g., mannitol) lead to widespread CNS uptake and potential off-target effects.

We also speculate that MRgFUS would continue to be the optimal approach compared to diffuse delivery across the BBB in cases where larger volumes, but still targeted drug delivery is desired, such as frontotemporal regions in ALS/FTD. Larger BBB opening volumes are possible by combining more targets than performed in this study, as well as sparser spacing of sonication spots as described previously (Lipsman et al. 2018). This is explained:

Lines 296-303: “While the minimal volume of BBB opening for sufficient therapeutic distribution is unknown and likely depends on drug properties, future trials will explore the safety and feasibility of sonicating larger volumes or the entirety of the primary motor cortex, along with frontotemporal regions that are also affected in ALS with frontotemporal dementia. Current preclinical evidence support the feasibility of targeting large hemispheric volumes in canines, whose brain dimensions more closely resemble that of humans.⁴⁶ Further, the clinical MRgFUS device can achieve a larger volume of BBB opening through a summation of targets and wider spacing of spots within each target.^{14”]}

There should be discussion regarding the most effective ways to look at potential injury at the site of treatment, beside the MR based measures (IR) and the clinical assessment to test features of upper motor neuron involvement following treatment. For example, would the use of DTI add anything to this assessment? The Authors provide mostly measures of clinical evaluation of global neurological impairment used in the functional rating of ALS (ALSFRSR), upper motor neuron function (spasticity scale) and cognitive function (MoCA) pre and post-treatment. These measures can lack sensitivity and are also examiner-dependent.

[Thank you for this suggestion. We agree that clinical measures lack sensitivity and reliability and imaging and neurophysiological biomarkers are desirable in future MRgFUS trials in ALS. However, candidate biomarkers have not been fully validated in ALS and biomarker development is a priority in the ALS field. We edited the revised discussion as follows:

Revised discussion:

Lines 325-328: “Neuroinflammation and endothelial, neuronal and glial function should be further investigated in upcoming MRgFUS trials using advanced MR imaging and

spectroscopy, cortical excitability measures (e.g., short intracortical interval parameter of threshold-tracking transcranial magnetic stimulation), PET imaging (e.g. TP50⁵¹), along with serum and cerebrospinal fluid biomarkers.”

We also measured the safety of BBB opening in the primary motor cortex using EEG, MRI and safety laboratory. We clarified these measures in the revised method and results. We further performed longitudinal structural/volumetric MRI analysis of the primary motor cortex, this new result is presented in Figure S3. The methods are described on line 446 – 452.

Revised methods:

Lines 528-531: “Safety laboratory testing included complete blood count, C-reactive protein, erythrocyte sedimentation rate, creatinine, cystatin-C, liver enzymes and electrolytes. Safety MRI sequences included T1 with and without gadolinium, T2, fluid-attenuated inversion recovery (FLAIR), T2 gradient echo (GRE), and diffusion weight imaging (DWI).”*

Revised results:

Lines 197-199: “In all participants, follow-up MR sequences did not demonstrate parenchymal or subarachnoid hemorrhage, ischemia, gliosis or worsened precentral gyrus cortical atrophy (Figure S3) up to 30 days following the intervention.”

Line 207: “All safety laboratory testing was unremarkable.”

We did consider DTI in the imaging protocol, but it was not included to balance the number of MR sequences necessary to assess safety and the time the ALS subjects could tolerate in the MRI suite. This sequence will be used in future phases of development.]

The Authors refer to previous published data about AAV9 introduction via lumbosacral intrathecal administration which nicely diffuse across multiple anterior horn levels of the spinal cord but has little diffusion into the motor cortex. This observation is clearly in favour of using MRgFUS for a more localised effect on BBB but would also raise the question to whether MRgFUS would work well in different neuroanatomic structures like in the spinal cord. Considering that lower motor neuron involvement remains a substantial aspect of the disease pathogenesis, what are the Author’s thoughts about the feasibility to use this technique for spinal cord BB modification?

[Thank you very much for this thoughtful comment. Considering the lower motor neuron involvement in ALS pathogenesis, we agree additional application of MRgFUS to the spinal cord may result in synergistic benefit for patients with ALS. We discussed potential MRgFUS blood-spinal cord barrier opening in the revised discussion as follows:

Lines 283-291: “To enhance a potential synergistic effect on UMNs and LMNs, targeted brain therapeutic delivery using MRgFUS can be coupled with non-targeted, lumbosacral intrathecal therapeutic injections, which have unreliable motor cortex penetration and are currently being used in ALS clinical trials. MRgFUS blood-spinal cord barrier opening has not yet been developed in humans, but has been trialled safely in the spinal cord of non-ALS rodent models^{41,42}. An early study demonstrated enhanced chemotherapy delivery using MRgFUS blood-spinal cord barrier opening in rodents.⁴³ However, the bony vertebrae of the spinal column in primates present a technical challenge for focusing the ultrasound and further validation and optimization studies are necessary before translation to humans.⁴⁴”

References:

- 42. Weber-Adrian, D. et al. Gene delivery to the spinal cord using MRI-guided focused ultrasound. *Gene Ther.* **22**, 568–577 (2015).*
- 43. Payne, A. H. et al. Magnetic resonance imaging-guided focused ultrasound to increase localized blood-spinal cord barrier permeability. *Neural Regen. Res.* **12**, 2045–2049 (2017).*
- 44. O’Reilly, M. A. et al. Preliminary Investigation of Focused Ultrasound-Facilitated Drug Delivery for the Treatment of Leptomeningeal Metastases. *Sci. Rep.* **8**, 9013 (2018).*
- 45. S. P. Fletcher & M. A. O’Reilly. Analysis of Multifrequency and Phase Keying Strategies for Focusing Ultrasound to the Human Vertebral Canal. *IEEE Trans. Ultrason. Ferroelectr. Freq. Control* **65**, 2322–2331 (2018).]*

Also ALS is notoriously a non-cell autonomous disease although, admittedly, the end point of the pathological process and the ultimate source of irreversible neurological dysfunction is motor cell loss. We know little about the origin of the pathological process and glia as well as axons may be deeply located. What is in the Authors opinion (also based on animal models studies and previous published data) the depth of tissue/BBB involvement which may condition the degree of penetration into cortical and sub-cortical tissue of a therapeutic agent? They refer in their manuscript to the aim of opening a maximum BBB volume of 1 cm.3

[The treatment envelope of MRgFUS for BBB opening covers essentially the entire brain: both superficial targets as described in this study and deeper targets as described in Lipsman et al. 2018 have been shown to be feasible and safe in human subjects. In animal models, the BBB of deeper structures (e.g. striatum and brainstem), has been successfully permeabilized using FUS. We refer to this now on line 139. Another advantage of MRgFUS is the procedure occurs at the microvascular level, improving the uniformity and depth of penetration of therapeutic delivery.]

In terms of target volume, animal studies showed that whole hemispheric volumes can be targeted with MRgFUS. Also, BBB opening of larger brain volumes, through multiple sonication volumes and a larger grid size (as in Lipsman et al. 2018), is an objective of currently ongoing clinical studies. This is discussed as a limitation to the study on line 284-287.]

The statement "...the lack of gadolinium enhancement in the brain and spinal cord of ALS subjects indicates a clinically impermeable BBB in ALS, which is in contrast to inflammatory CNS diseases such as multiple sclerosis (MS) which are characterized by gadolinium enhancement...". It would be important also to note that there are numerous reports suggesting a BBB damage in ALS and that this has been identified using different means of investigation which are not necessarily dependent on gadolinium enhancement. Therefore, it is difficult to be absolutely sure about the degree of BBB permeability in a pathological condition like ALS.

[Thank you again for this thoughtful comment. As informed by a systematic review on BBB abnormalities in ALS conducted by our team (registered at Prospero, http://www.crd.york.ac.uk/PROSPERO/display_record.php?ID=CRD42017065405), we agree there are a number of preclinical and post-mortem studies showing BBB abnormalities in ALS. This was acknowledged in our manuscript (lines 110-111: "While there is preclinical and post-mortem evidence of abnormal BBB ultrastructure in ALS^{7,8}".

In the revised manuscript, we decided to remove this statement of gadolinium enhancement in ALS to avoid misleading inferences.

References:

- 7. Garbuzova-Davis, S. et al. Ultrastructure of blood-brain barrier and blood-spinal cord barrier in SOD1 mice modeling ALS. Brain Res 1157, 126–37 (2007).*
- 8. Garbuzova-Davis, S. & Sanberg, P. R. Blood-CNS Barrier Impairment in ALS patients versus an animal model. Front Cell Neurosci 8, 21 (2014).]*

The Authors emphasize the theory of an upper motor neuron dominant origin of ALS, which would justify in full the use of their technique for transitory permeabilization of the BBB. I think it may be worthwhile also mention the theory of distal axonopathy, which includes the concept of neuromuscular junction primary involvement and a caudal to rostral progression of the disease

*[Thanks for highlighting this point that improved our revised introduction as follows:
Lines 99-101: " While preclinical studies in ALS rodent models have primarily focused on the distal-to-proximal motor neuron dying-back hypothesis¹, recent data support a key pathological role of UMN in the motor cortex.²"]*

I would also like to refer to the statement “..Longitudinal neurological assessments revealed no significant change in disease progression as indicated by ALSFRS-R and MRC scores for the limb contralateral to the sonicated cortex...”

I think the Authors may want to say that in their view, there was no procedure-induced acceleration of disease progression, as the disease would be expected to progress over time (we would not know whether BBB permeabilization in itself has a positive disease modifying effect). I think it may be worth specifying what is the length of longitudinal follow-up over time they are referring to and state what parameter to evaluate disease progression they have used. If this is ALSFRSR, then what may be useful is to plot the ALSFRSR temporal slope pre and post treatment to show the relative steady progression of the disease or the lack of progression in the pre to post-procedure follow-up. The Authors have accurately selected their ALS subjects, with a range of disease durations to time of treatment of 4 to 4.5 years (with an average ALSFRSR at treatment of 30 to 35), which is likely to be a stage 2 or 3 of the disease, and modified Ashworth score of 0 at the time of treatment (to allow for a better characterization of potential cortico-spinal / upper motor neuron changes after treatment).

[Thank you for your positive comments regarding our trial design and enrollment criteria.

We reworded this sentence according to your suggestions:

Lines 200-204: “Longitudinal neurological assessments revealed no accelerated disease progression as measured by clinical assessment and MRC scores for the limb contralateral to the sonicated cortex up to 30 days after sonications and ALSFRS-R scores up to day 60 (Table S3).”

We provided the ALSFRS-R scores in Table S3 for the readers.

Once again, many thanks for taking the time to review our study and for your supportive comments. We truly appreciate your suggestions that have significantly improved our manuscript.]

REVIEWERS' COMMENTS:

Reviewer #1 (Remarks to the Author):

very nice research. ok for publication

Reviewer #2 (Remarks to the Author):

The Authors have addressed comprehensively the points raised in my review of the submitted article. While it is objectively difficult to set a standard of the most informative procedures that may be required to evaluate the effects of MRgFUS for targeted treatment delivery, I believe that the monitoring of the downstream effects of treatment is robust in the evaluation of clinical and biological consequences of the procedure. This is one of the first in-human attempt to use this method to pave the way to the development of affective treatment strategies for ALS, a devastating neurodegenerative disorder. Clearly the main issue here remains how this approach will be used taking into account the balance between a targeted BBB permeabilisation for treatment delivery to hot spots of disease and the fact that ALS may be to a different degree, a widespread pathological process. Hence, it is not clear how to define targets and more importantly, how not to be "off target". Nevertheless, I think that the data the Authors present are robust and they have followed a good rationale in their experimental plan. They have now improved the paper with a thorough analysis of the potential limitations of this technique and elaborated on what could be done next to expand their initial encouraging results. I would definitely encourage publication at this stage.

Response Letter

REVIEWERS' COMMENTS:

Reviewer #1 (Remarks to the Author):

very nice research. ok for publication

Once again, many thanks for taking the time to review our study and for your supportive comment. We truly appreciated your suggestions that have significantly improved our manuscript.

Reviewer #2 (Remarks to the Author):

The Authors have addressed comprehensively the points raised in my review of the submitted article. While it is objectively difficult to set a standard of the most informative procedures that may be required to evaluate the effects of MRgFUS for targeted treatment delivery, I believe that the monitoring of the downstream effects of treatment is robust in the evaluation of clinical and biological consequences of the procedure. This is one of the first in-human attempt to use this method to pave the way to the development of affective treatment strategies for ALS, a devastating neurodegenerative disorder. Clearly the main issue here remains how this approach will be used taking into account the balance between a targeted BBB permeabilisation for treatment delivery to hot spots of disease and the fact that ALS may be to a different

degree, a widespread pathological process. Hence, it is not clear how to define targets and more importantly, how not to be "off target". Nevertheless, I think that the data the Authors present are robust and they have followed a good rationale in their experimental plan. They have now improved the paper with a thorough analysis of the potential limitations of this technique and elaborated on what could be done next to expand their initial encouraging results. I would definitely encourage publication at this stage.

Thank you for your thoughtful comments and for encouraging the Editorial team to publish our revised manuscript.